# The Multifunctionally Graded System for a Controlled Size Effect on Iron Oxide–Gold Based Core-Shell Nanoparticles

**DOI:** 10.3390/nano11071695

**Published:** 2021-06-28

**Authors:** Bo-Wei Du, Chih-Yuan Chu, Ching-Chang Lin, Fu-Hsiang Ko

**Affiliations:** 1Department of Materials Science and Engineering, National Yang Ming Chiao Tung University, Hsinchu 30010, Taiwan; duu.mse04g@nctu.edu.tw (B.-W.D.); dv8962@hotmail.com (C.-Y.C.); 2Research Center for Advanced Science and Technology (RCAST), The University of Tokyo, 4-6-1 Komaba, Meguro-ku, Tokyo 153-8904, Japan; Lin@dsc.rcast.u-tokyo.ac.jp

**Keywords:** molecular carriers, magnetic nanoparticles, core-shell nanostructure, aptamer, size effect

## Abstract

We report that Fe_3_O_4_@Au core-shell nanoparticles (NPs) serve as a multifunctional molecule delivery platform. This platform is also suitable for sensing the doxorubicin (DOX) through DNA hybridization, and the amount of carried DOX molecules was determined by size-dependent Fe_3_O_4_@Au NPs. The limits of detection (LODs) for DOX was found to be 1.839 nM. In our approach, an Au nano-shell coating was coupled with a specially designed DNA sequence using thiol bonding. By means of a high-frequency magnetic field (HFMF), a high release percentage of such a molecule could be efficiently achieved in a relatively short period of time. Furthermore, the thickness increase of the Au nano-shell affords Fe_3_O_4_@Au NPs with a larger surface area and a smaller temperature increment due to shielding effects from magnetic field. The change of magnetic property may enable the developed Fe_3_O_4_@Au-dsDNA/DOX NPs to be used as future nanocarrier material. More importantly, the core-shell NP structures were demonstrated to act as a controllable and efficient factor for molecule delivery.

## 1. Introduction

The past four decades have seen the foundations established for nanotechnologies to deliver therapeutic and diagnostic agents in one securer and more effective manner [1,2,3,4]. With the development of nanotechnology, various nanoparticles (NPs), nanocarriers, or conjugates have been developed for various biomedical applications. Owing to their specific configurational properties and favorable physical–chemical characteristics [5,6], the goal of modulating both the pharmacokinetic and pharmacodynamic profiles of drugs can be achieved to enhance their therapeutic index [7,8,9]. In particular, inorganic NPs have gradually become more popular in recent years. Integration of magnetic iron oxide (Fe_3_O_4_) and Au NPs can be used in hyperthermia [10], catalysis [11], and surface modification [12]. Furthermore, Fe_3_O_4_ NPs have been used as carriers for cell imaging [13] and drug delivery systems [14] due to their apparent superparamagnetism, high specific surface area, significant colloidal stability, and excellent biocompatibility [15,16]. An Au coating onto magnetic NPs is another appealing hybrid system [17,18]. Except for single-component materials, the multicomponent materials demonstrate unique property. With an Au coating, magnetic NPs can be functionalized through thiol linkages. The Au coating also renders the magnetic NPs with a surface plasmon resonance (SPR) effect such that conduction electrons have resonant oscillation at the interface between signed permittivity materials when exposed to incident light. Through proper selection of the particle size, Au colloidal nanospheres display strong SPR absorption intensity in the near-infrared (NIR) region [19], enabling their use for computed tomography (CT) imaging [20] and magnetic resonance imaging (MRI) [21] of cancer cells or other biological systems.

For cancer diagnosis and medical treatment, it is necessary to devise a therapy that is capable of integrating both target drug delivery systems into cancer cells and that has fewer effects on normal cells. In particular, for the fewest side effects, it is crucial to focus the drug on the tissues of interest in the meantime decreasing the relative concentration of the medication in other tissues so that molecule that can specifically bind to the target cell [22,23]. For instance, once antibodies link with antigens and the antibody Fc domains engage Fc receptors on the immune effector cell surfaces [24], the antibody-dependent cellular cytotoxicity appears, which triggers the immune system to kill cancer cells after antibodies bind to antigens [25]. Antibodies are also controlled by complement-dependent cytotoxicity, which is another cell-killing technique [26]. Aptamers, as artificial nucleic acid ligands formed forward specific targets, have testified to a novel kind of ligands that rival antibodies in their potential for therapeutic and diagnostic applications [27,28]. Alternatively, aptamers provide more attractive properties on therapeutic agents than antibodies [29]. For example, aptamers have better stability and can stand for much more extreme environments, such as high temperatures [30,31]. In addition, due to their better diversity in binding targets, aptamers can be designed not only for targeting but also for functioning as drug carriers [32]. Previous studies have also indicated that aptamers are absent of immunogenicity and can be chemically modified to enhance the performance toward nucleases or lengthen the period of their blood circulation [33]. In addition to these properties of the materials, NP size is a vital characteristic that has been indicated to have a great effect on biomedical applications [34,35]. In addition to the SPR effect of NPs as mentioned, previous reports showed that magnetic NPs of different diameters corresponded with different MRI signals [36], and this effect was attributed to the change in the NP surface area. Jiang et al. showed the relationship between the NP diameter and the amount of protein that could be coupled on the NP surface [37]. Therefore, for accurate theranostics of cancer cells or targeting-type drug delivery systems, it is essential to discuss the size effect of nanoparticles, and the effect that may be related to the shielding effect on magnetic property [38,39,40].

In the present work, we reported the synthesis of Fe_3_O_4_@Au core-shell NPs, which serve as substrates for further application in multifunctional molecule delivery systems. Both magnetic and optical properties derived from the Fe_3_O_4_ NPs and the Au nano-shells were studies. The specially designed CG-rich fragments by triggering with HFMF as the purpose of the actuator, and the designed Fe_3_O_4_@Au-dsDNA material can carry DOX (as the anticancer drug). Hence, both molecule loading and specific target application could be achieved (Scheme 1) [41]. In addition, we synthesized Fe_3_O_4_@Au NPs material with various size distributions by adjusting the concentration of the reduction agent. The different Au nano-shell thicknesses were studied with respect to the magnetism, surface area, and other physical functions. Such an approach is beneficial to understanding the basic property of our molecule carrier for future targeted drug delivery and cancer therapy. The formation of a larger surface area, which leads to a higher DNA binding concentration, carries more DOX molecules and results in a difference in chemotherapeutic ability. The cytotoxicity of ds-DNA-conjugated Fe_3_O_4_@Au drug carriers was investigated with respective to cell death and biocompatibility of our carriers.

## 2. Materials and Methods

### 2.1. Materials

Iron(II) chloride tetrahydrate (FeCl_2_·4H_2_O, 98%), iron(III) chloride hexahydrate (FeCl_3_·6H_2_O, 98%), and hydroxylamine hydrochloride (NH_2_OH·HCl, 99%) were purchased from Alfa Aesar (Lancashire, UK). Hydrogen chloride (HCl, ≥99%), Au(III) chloride trihydrate (HAuCl_3_·H_2_O, ≥49%), doxorubicin hydrochloride (DOX, C_27_H_29_NO·HCl, ≥ 98%), phosphate-buffered saline (PBS, 99.18%), tetramethylammonium hydroxide (TMAOH, ≥97%) and dimethyl sulfoxide (DMSO, ≥99.9%) were purchased from Sigma (MO, USA). Sodium hydroxide (NaOH, 97%) and trisodium citrate dihydrate (C_6_H_5_Na_3_O_7_·2H_2_O, 98%) were obtained from SHOWA (Tokyo, Japan). Tetramethylammonium hydroxide ((CH_3_)_4_NOH, 25% *w/w* in aqueous solution) was used as purchased from Fluka (Heidelberg, Germany). Dulbecco’s modified Eagle’s medium (DMEM) was purchased from Biowest (MO, USA). Trypsin and tetrazolium salt 3-(4,5-dimethylthiazol-2-yl)-2,5-diphenyltetrazolium bromide (MTT, ≥99.9%) were purchased from Thermo Scientific (MA, USA) and Bersing Bioscience Technology (North District, Taiwan), respectively. Oligonucleotides were obtained from MDBio Inc. (Taipei, Taiwan) and the target DNA containing the 5′-AAAAAAAAAAAAAAATCGTCGTCGTCGTCGTCGAAAAAAAAAAAAAAAAAGCAGTTGATCGTTTGGATACCCTGG-3′ sequence and the complementary DNA containing the 3′-TTTAGCAGCAGCAGCAGCAGCTT-5′ sequence were hybridized to form double-strand DNA. A Milli-Q deionized water system provided deionized water in all processes, which with a resistivity more than 18 MΩ·cm. HeLa cells (a human cervical carcinoma cell line; Sigma, MO, USA) and MCF-7 (Sigma, MO, USA) cells were used for the cell viability test.

### 2.2. Preparation of Superparamagnetic Fe_3_O_4_ NPs

The Fe_3_O_4_ NPs were prepared in an aqueous solution using the coprecipitation method [42] according to the following procedure: 5.4 g (20 mmol) FeCl_3_·6H_2_O followed by 2.0 g (10 mmol) FeCl_2_·4H_2_O were dissolved in 25 mL of 6 M HCl (aq). The solution was added dropwise into 25 mL of 1.5 M NaOH solution with energetic stirring, and the black precipitates were isolated by a magnet and washed twice with both deionized water and 0.1 M tetramethylammonium hydroxide pentahydrate. The particles were separated by centrifugation at 14,000 rpm for 30 min, meanwhile, the ultimate product was dispersed in 250 mL of 0.1 M TMAOH. The 6.5 mg/mL magnetic NP solution was stored at room temperature under benchtop conditions.

### 2.3. Preparation of Fe_3_O_4_@Au NPs

Fe_3_O_4_@Au core-shell NPs were synthesized by deposition of Au on the performed Fe_3_O_4_ NPs using a modification of Lyon’s iterative hydroxylamine seeding procedure in three steps. First, 1 mL of the Fe_3_O_4_ NP solution was mixed by vortexing with 1 mL of 0.1 M sodium citrate for 10 min. Next, 20 mL of deionized water added into the solution, and then 100 μL of 80 mM NH_2_OH·HCl was added to this solution. Finally, 2 mL of 1% HAuCl_4_ solution was added dropwise with stirring. The uncoated Fe_3_O_4_ NPs were removed by centrifugation at 6000 rpm for 5 min.

Size distribution of Fe_3_O_4_@Au core-shell NPs was synthesized by alternating the amount of sodium citrate and NH_2_OH·HCl. The conditions of each NP size are shown below (Table 1).

### 2.4. Hybridization of Double-Stranded Oligonucleotides (dsDNA)

The single-stranded DNA (ssDNA) was added to TE buffer (10 mM Tris-HCl/1 mM EDTA)/50 mM NaCl with pipetting to form an ssDNA solution. A15T3 dsDNA was hybridized with A15 ssDNA and T3 ssDNA sequences at a 1:1 molar ratio in a large water bath heated to 95 °C for 10 min and then cooled down to room temperature. The obtained dsDNA was stored at −20 °C.

### 2.5. Preparation of F Fe_3_O_4_@Au NPs-dsDNA and DOX-Intercalated Fe_3_O_4_@Au NPs-dsDNA

One milliliter of the 0.1 mg/mL Fe_3_O_4_@Au NP solution was mixed with 10 μL of the 50 mM dsDNA at room temperature for 6 h. After being centrifuged, the residual product was washed twice to remove unbound dsDNA and then resuspended in 1 mL of deionized water.

For DOX molecule loading, 1 mL of the 0.1 mg/mL Fe_3_O_4_@Au NPs-dsDNA was mixed with 5 μL of 1.72 mM DOX at room temperature for 3 h. After the reaction, the solution was centrifuged at 11,500 rpm for 15 min to remove unintercalated DOX, and the pellet was resuspended in 1 mL of deionized water.

### 2.6. Characterization of the NPs

The morphology and structure of the Fe_3_O_4_ and Fe_3_O_4_@Au NPs were determined by transmission electron microscopy (TEM, JEOL, JEM-2010, Akishima, Japan) and scanning electron microscopy (FE-SEM, JEOL-6700, Akishima, Japan). UV-Vis spectroscopy (HITACHI, U-3310, Tokyo, Japan) was used to determine the Au nano-shell coverage and the size of the Fe_3_O_4_@Au NPs. Energy-dispersive X-ray spectroscopy (Oxford-Link ISIS 300 energy-dispersive X-ray, High Wycombe, UK) was used to analyze the NP elemental distribution under TEM. X-ray diffraction signals of the NPs were obtained by X-ray diffraction (XRD, X’Pert PRO MRD system, Almelo, Netherlands). Fluorescence spectroscopy was used to measure the DOX released from different Fe_3_O_4_@Au NP carriers using a fluorescence spectrophotometer (HITACHI, F-7000, Tokyo, Japan). The superconducting quantum interface device vibrating sample magnetometer (SQUID, MPMS-XL, CA, USA) was used for estimating the magnetic properties of the NPs. The high-frequency magnetic field (HFMF) with a frequency of 50 kHz and a magnetic field strength (H) of 8 kA/m was used for providing an oscillating magnetic field to heat the magnetic materials.

### 2.7. Molecule Release by Diffusion and under HFMF

For testing the DOX molecule release, 1 mL 0.1 mg/mL of Fe_3_O_4_@Au NPs-dsDNA/DOX was placed in the 37 °C water bath, DOX was released slowly from dsDNA. The samples were taken out of the water bath at different times and centrifuged to collect the supernatant immediately.

Each size of Fe_3_O_4_@Au NPs-dsDNA/DOX was put in the center of the loop of HFMF without any contact over different time periods. After treating with HFMF, the sample was removed immediately, and the released molecules were separated with nanoparticles through centrifugation at 14,000 rpm for 30 min.

### 2.8. In Vitro Cytotoxicity Assay of Fe_3_O_4_@Au NPs-dsDNA and Fe_3_O_4_@Au NPs-dsDNA/DOX

HeLa cells were cultured in DMEM supplemented with 10% fetal bovine serum and 1% penicillin–streptomycin and then incubated in humidified air containing a 5% CO_2_ atmosphere at 37 °C and changed every 2.5 d. Fe_3_O_4_@Au NPs, Fe_3_O_4_@Au NPs-dsDNA, and Fe_3_O_4_@Au NPs-dsDNA/DOX of each size were prepared, and Fe_3_O_4_@Au NPs were incubated with Tween 20 (0.05%) to avoid aggregation in the medium. Subsequently, NPs were dissolved in DMEM after centrifugation at 6000 rpm for 5 min.

After detached with trypsin, 5000 cells were cultured in 96-well plates with 200 μL DMEM for 24 h to assign the cells to be linked with the plate. The medium was supplemented with 1% penicillin–streptomycin and without fetal bovine serum in DMEM. After 24 h of incubation, the DMEM suspension containing the NPs (0.1 mg/mL) was substituted for the medium, and the cells were further incubated for 24 h. Subsequently, 100 μL of the medium was eliminated, and 10 μL of the 5 mg/mL MTT was added for another 4 h of incubation. Then, all solutions were removed and replaced with 100 μL DMSO solution. The quantification of cell situation was observed using an ELISA plate reader at a wavelength of 570 nm.

### 2.9. Size Effects on Molecule Delivery under HFMF

HeLa cells were cultured in DMEM added in 10% fetal bovine serum and 1% penicillin–streptomycin. Then, the cells were incubated in humidified air containing a 5% CO_2_ atmosphere at 37 °C and changed every 2.5 d. Each Fe_3_O_4_@Au NPs-dsDNA/DOX (0.1 mg/mL) size was dissolved in DMEM after centrifugation at 6000 rpm for 5 min. After being detached with trypsin, 5000 cells were added to 200 μL DMEM containing Fe_3_O_4_@Au NPs-dsDNA/DOX in an Eppendorf tube. The Eppendorf tube was placed in the center of the loop of the HFMF without any contact for 7 min, and then the tube was removed immediately. DMEM with Fe_3_O_4_@Au NPs-dsDNA/DOX and the cells were moved into a well of a 96-well plate and cultured for 24 h. Then, the 100 μL of the medium was removed, and 10 μL of 5 mg/mL MTT was added for another 4 h of incubation. Then, all the solutions were removed and replaced with 100 μL DMSO solution. The cells were quantified using an ELISA plate reader at a wavelength of 570 nm.

### 2.10. Target Molecule Delivery under HFMF

HeLa and MCF-7 cells were cultured in DMEM supplemented with 10% fetal bovine serum and 1% penicillin–streptomycin. Then, the cells were incubated in humidified air containing a 5% CO_2_ atmosphere at 37 °C and changed every 2.5 d. After being detached with trypsin, 90,000 cells were plated in each well of a 6-well plate with a total of 3 mL DMEM (0.1 mg/mL Fe_3_O_4_@Au NPs-dsDNA/DOX) supplemented with 10% fetal bovine serum and 1% penicillin–streptomycin and then placed in an incubator for another 24 h. The medium was changed to DMEM supplemented with 1% penicillin–streptomycin with no fetal bovine serum. After incubation for 24 h, 2 mL of the medium with the different sized Fe_3_O_4_@Au NPs-dsDNA/DOX were substituted for the original medium. Before washing with PBS buffer, all plates were placed in an incubator for 4 h for targeting. After cells were subcultured from each other and treated with HFMF, the cells were seeded into 96-well plates at 500 cells per well and further cultured for one day. To determine cell viability, the MTT assay was directly performed.

## 3. Results and Discussion

### 3.1. Synthesis and Characterization of Fe_3_O_4_@Au NPs

The Fe_3_O_4_ NPs were successfully synthesized by the coprecipitation method and can be well dispersed in an aqueous solution of TMAOH. In contrast to poor dispersion in deionized water (see Figure 1a), the TMAOH is a proper surfactant for stability and better solution dispersion. Figure 1b shows that the size of the Fe_3_O_4_ NPs is approximately 10 nm. Subsequently, the Fe_3_O_4_ NPs were covered with an Au shell by reducing HAuCl_4_ with sodium citrate. Scanning electron microscopy images (Figure 1a (right) and Figure 1c) were taken to calculate the size distribution (Image-Pro Plus, MD, USA), and the average sizes of the Fe_3_O_4_ and Fe_3_O_4_@Au NPs were 9.6 ± 3.0 nm and 39.5 ± 3.0 nm, respectively (Figure 1g,h). Figure 1d shows that the NPs exhibited a core-shell morphology. And the NPs showed an average size of 40 nm without any aggregation, which corresponded to the aforementioned SEM image (Figure 1c). The selected area electron diffraction (SAED) patterns were also used to study the core-shell structure of the NPs. According to the SAED patterns, Fe_3_O_4_ NPs can be indexed as Fe_3_O_4_ (JCPD 89-6446) corresponding to (200), (230), and (133); the Fe_3_O_4_@Au NPs can be indexed as Fe_3_O_4_ (JCPD 89-6446) corresponding to (122) and (006), and the (311) and (220) planes were referred to Au (JCPD 89-3697 and 65-2870) (Figure 1e). Due to the SPR of the Au shell, UV-Vis spectroscopy was used to confirm the existence of the Au shell. A strong absorbance peak at 530 nm in the Fe_3_O_4_@Au NPs was absent in the spectrum of pure Fe_3_O_4_ NPs (Figure 1f).

To discuss the influence of different Fe_3_O_4_@Au NP sizes, the SEM images and size distribution calculated by Image Pro-Plus (IPP) software were studied at different reactant concentrations (Figure 2). According to the SEM images, the size distribution of the Fe_3_O_4_@Au NPs exhibited mean diameters of 59.5 ± 10.5 nm, 47.7 ± 7.1 nm, 39.5 ± 5.7 nm, 33.1 ± 5.0 nm, and 25.9 ± 6.0 nm. According to the average size of Fe_3_O_4_@Au NPs, the gold nano-shell sizes distribution was shown: 49.9 ± 7.5 nm, 38.1 ± 4.1 nm, 29.9 ± 2.7 nm, 23.5 ± 2.0 nm, and 16.3 ± 3.0 nm, respectively. Since the size distribution of NPs has a large impact on the absorbance peak caused by the surface plasma resonance effect, the absorbance peak at approximately 530 nm was shown to be the characteristic peak of the Au NPs (Figure 3a). For different Fe_3_O_4_@Au NP sizes, the inset Figure shows that the absorbance peaks of larger NPs had a greater redshift than those of smaller NPs, which shows that the Au NP size was closely related to their optical characteristics. The particle sizes decreased from 59.5 to 25.9 nm, and absorbance peaks could be observed at 535, 532, 530, 528, and 525 nm. The crystalline structure of the Fe_3_O_4_@Au NPs of different sizes was characterized by XRD (Figure 3b). The diffraction peaks of the Fe_3_O_4_@Au NPs are indicated at 38.2°, 44.4°, 64.6°, and 77.6°, which can be indexed to the (111), (200), (220), and (311) planes of Au in a cubic phase with a JCPD code of 65-2870. Furthermore, the intensity of the (311) plane of the Fe_3_O_4_ NPs decreased as the thickness of the Au nano-shell increased. In addition, the increase in each Au diffraction peak with increasing particle size confirmed that the composition of the Au increased and presented that the Fe_3_O_4_ NPs were fully covered with Au nano-shells without Fe_3_O_4_ NPs. The magnetic properties of the Fe_3_O_4_@Au NPs were investigated by SQUID (at 25 °C with the magnetic field sweeping from −18,000 to +18,000 G). As shown in Figure 3c, all curves for the Fe_3_O_4_ NPs and Fe_3_O_4_@Au NPs had similar shapes with negligible hysteresis, signifying superparamagnetic properties. The saturation magnetization (Ms) of the Fe_3_O_4_ NPs was 25 emu/g, and the value decreased gradually with increasing Au nano-shell thickness. The relation between the Au nano-shell thickness and the saturation magnetization value (Figure 3d) was quantified. The linear relation between the ratio of Fe_3_O_4_ NPs to Fe_3_O_4_@Au NPs and the saturation magnetization value is *y* = 6.02*x* + 0.6 with a correlation coefficient of 0.98. The Au nano-shell in this study provided an increasing shielding effect for the Fe_3_O_4_ NPs as the thickness of the Au nano-shell increased.

### 3.2. Quantitative Analysis on Molecule Loading and Sensing Capacity

The nanocarriers were fabricated by Fe_3_O_4_@Au NPs-dsDNA/DOX, to quantify the anticancer effect of DOX, the detection limits (LODs) calculation towards DOX was performed through standard deviation and linear fittings and DOX calibration curves of (*y* = 528.19*x* + 0.35) were obtained for the PL spectrum (Appendix A). A previous study indicated that DOX intercalated prior into consecutive CG base pairs; as a consequence, the related oligonucleotide C(GA)_6_, which enhanced the DOX-loading capacity, provided binding sites for a minimum of 6 DOX molecules. Since the fluorescence of DOX was quenched when intercalated into oligonucleotides, the calibration curve (*y* = 9951.28 − 1367.19*x* × ln(*x* + 4.72)) was defined as the ability to bind DOX with DNA (Appendix A). The nonlinear relation between the remaining DOX intensity and the concentration of DNA, which also illustrated the nonlinear quenching ability of DNA, is also used to calculate the binding constant of the drug-DNA complex [43]. It is also important to provide the number of DOX molecules bound by each single DNA strand by plotting the fluorescence intensity (log[(F_0_ − F)/F]) and various concentrations of DNA (log[DNA]) (Appendix A). The results showed a slope value of 0.9, which indicates that 0.9 DOX molecules were bound by each single DNA strand, and the binding constant (K) of DOX molecules and DNA was 0.026. This curve was also used to further calculate the concentration of DNA binding on various Fe_3_O_4_@Au NP sizes. To further identify the concentration of DOX that had intercalated into oligonucleotides, the remaining concentration minus the original concentration was defined as the different NP sizes (26.0, 33.1, 39.5, 47.7, and 59.5 nm) intercalated with 4.6, 4.0, 3.6, 3.9, and 4.3 μM DOX. The concentrations of DNA that bound to different NP sizes were 379.8, 309.9, 274.0, 304.0, and 346.6 nM, respectively, because of the electrostatic force between the Fe_3_O_4_@Au NP and DNA. Furthermore, the zeta potential was applied to confirm the DNA sequences and intercalation of DOX. Table 2 suggests that the zeta potential decreased with increasing Fe_3_O_4_@Au NP diameter. After conjugation with DNA, the zeta potential slightly increased due to the negative charge of the DNA molecules [44], and a greater DNA binding concentration resulted in a greater zeta potential value. In addition, intercalated DOX caused a dramatic reduction due to the interaction between the opposite charges of DOX and DNA molecules [45]. We speculated that the surface area may strongly influence the oligonucleotide binding concentration. Figure 4a,b shows the relationship between the DNA and DOX concentration and the Fe_3_O_4_@Au NP surface area. These results could not be represented by the trend line due to the difference of particle numbers between each size of Fe_3_O_4_@Au NPs. After excluding the effect of particle number, the binding concentration of oligonucleotide per particle linearly correlate with the oligonucleotide binding concentration per particle surface area, the linear relation is *y* = 4.34 × 10^−14^*x* − 2.43 × 10^−10^ (*R*^2^ = 0.98) and *y* = 5.43 × 10^−16^*x* − 2.89 × 10^−12^ (*R*^2^ = 0.99), respectively (Figure 4c,d). Our result suggests that the surface area is a crucial factor that influences the oligonucleotide binding concentration and the DOX molecule loading capability.

### 3.3. Capabilities of the Multifunctional Molecule Delivery System

HFMF was used to induce hyperthermia, which is a therapy for cancer cells with heat generated by magnetic NPs under a strong oscillating magnetic field. After a slight temperature increase (at approximately 40 °C), a series of subcellular events are initiated and give rise to the cells susceptible to diverse damage, resulting in subsequent cell death. Moreover, increasing the temperature can also increase the thermosensitivity of cancer cells, which helps improve the efficiency of chemotherapy [46,47]. It is obvious that the Fe_3_O_4_@Au NPs with thinner Au nano-shells lead to a more conspicuous temperature rise, and the solution temperature attains 39.8 °C at a Fe_3_O_4_@Au NP diameter of 25.9 nm after a magnetic field is applied for 20 min (Figure 5a). By contrast, the largest Fe_3_O_4_@Au NPs (diameter of 59.5 nm) increased to 38.4 °C under the same experimental conditions. The specific loss power (SLP) was used to represent the ability of thermogenesis of the NPs from the magnetic coupling between the magnetic moment of the NPs and the utilized HFMF and was defined as the thermal power dissipation divided by the mass of magnetic NPs. The value of the SLP was applied to more accurately represent the ability of the NPs to be used in hyperthermia applications and was calculated by Equation (1), where ΔT is the vibrational temperature, Δt is the period of time, mf is the weight of magnetic NPs, C_f_ is the heat capacity of Fe_3_O_4_ NPs, mw is the weight of the solution (deionized water), and C_w_ is the heat capacity of water [48]. As shown in Table 3, the SLP value decreased from 148.95 to 91.24 w/g when the diameter of the Fe_3_O_4_@Au NPs increased from 25.92 to 59.50 nm. The results indicate that a thicker Au shell causes a lower magnetically induced heating ability.
(1)$$\mathrm{SLP}=\frac{\Delta T}{\Delta t}\times \frac{m_{f}C_{f}+{\;m}_{w}C_{w}}{m_{f}}$$

To further understand the stability of the molecule delivery system, the release percentage was measured at 37 °C (in physiological conditions). The plot of release percentage (%) in 70 h versus the diameter of the Fe_3_O_4_@Au NPs showed the same maximum release percentage of approximately 30–40% for each Fe_3_O_4_@Au NP size. This percentage is acceptable stability for DOX molecule delivery (Figure 5b). The release percentage showed a significant decrease after 24 h, which may be caused by a decrease in the fluorescence after a long period of time or the re-intercalation of DOX into oligonucleotides. In contrast, as a result of the heat produced by Fe_3_O_4_@Au NPs under HFMF, DOX that had intercalated in oligonucleotides could be released. With increasing time (from 0 to 20 min), the release percentage increased to 60–80% for different Fe_3_O_4_@Au NP diameters (from 25.9 to 59.5 nm) at 37 °C (Figure 5c), the increase rate was clearly slower than the first 7 min, as the reason for the applying time of HFMF. To further illustrate the relationship between the particle size and the DOX molecule release ability, the DOX concentration released per particle (μM) was plotted as a function of the DOX concentration carried per particle (μM) (Appendix A): a linear relation between the released and carried concentration (*y* = 0.7691*x −* 8.2156, *R*^2^ = 0.99) was found.

### 3.4. Application of Release Actuator for Treatment

We then investigated the integrative ability as a multifunctional DOX molecule delivery system using an in vitro test. In this study, the cytotoxicity of our system was the main concern before further applications. The MTT assay was performed to test the targeting ability of the molecule delivery system through the observation of the absorbance to assess the number of cells that survive. Two different cell lines (HeLa and MCF-7) have been used to compare the cell viability. Especially, the MCF-7 was chosen as a targeted cell due to its specific binding ability with the aptamer, which the HeLa cells do not have. We selected the HeLa cell line as a host cell to investigate the cytotoxicity of different Fe_3_O_4_@Au NP sizes and Fe_3_O_4_@Au NPs-dsDNA with Fe_3_O_4_@Au NPs-dsDNA/DOX as a negative control group. Among these three systems, the Fe_3_O_4_@Au NPs were covered with Tween 20 as a protective agent to prevent the aggregation of NPs when suspended in a culture medium. Since the dsDNA conjugated on the surface of the Fe_3_O_4_@Au NPs could also be applied as a protection agent, these two systems were examined without adding Tween 20. After incubation with HeLa cells with differently sized NP systems for 24 h, the cell viability remained approximately 100% or even higher after incubation with Fe_3_O_4_@Au NPs (Figure 6a). Fe_3_O_4_@Au NP-dsDNA, with a particle size of 39.5 nm, showed the lowest cell viability. Compared with a particle size of 25.9 nm, which was the largest, the viability showed a similar trend as the DNA binding concentration, which indicates that the DNA binding concentration could have a great influence on cell viability. Since DNA is a biocompatible molecule, it could offer lower cytotoxicity for our system; nevertheless, even though the cell viability showed different values for each NP size, the lowest cell viability was still greater than 90%. In addition, DOX-loaded Fe_3_O_4_@Au NPs-dsDNA showed lower cell viability for each size compared to Fe_3_O_4_@Au NPs-dsDNA without DOX molecule loading, and the results were caused by the diffusive release of DOX from the substrate. This can be further confirmed by the Fe_3_O_4_@Au NPs-dsDNA with a particle size of 39.5 nm having the lowest DOX loading concentration and the greatest cell viability among other particle sizes. The ability as a cancer therapy agent was examined by mixing HeLa cells with different of Fe_3_O_4_@Au NPs-dsDNA/DOX sizes and applying HFMF for 7 min. After the magnetic field treatment, the cells were further incubated for 12 h for proper attachment before the MTT assay. Importantly, compared to the control group, the cell viability of the DOX-loaded Fe_3_O_4_@Au NP-dsDNA group was significantly lower, showing that our system has the good capability as a cancer agent. For pure DOX, 8.6 μM caused approximately 75% endocytosis, whereas Fe_3_O_4_@Au NPs-dsDNA/DOX achieved the same effect with only 2 μM DOX release.

Except for lower DOX concentration, targeting the carrier to a specific cell is an efficient way to decrease side effects, and MCF-7, a targeted cell, was employed in this study. An aptamer sequence that had been demonstrated to have a specific binding ability with MCF-7 breast cancer cells was chosen [49], and HeLa cells were used as the control group, nanoparticles could be easily washed off by PBS. There was a notable distinction between the HeLa and MCF-7 cells after HFMF treatment; for the HeLa cells, cell viability remained greater than 95% among all DOX molecule carriers (Fe_3_O_4_@Au NPs-dsDNA/DOX) of different sizes (Figure 6b). In contrast, the cell viability for MCF-7 cells decreased due to the effects of hyperthermia and chemotherapy (Figure 6c,d). Moreover, the IC50 was also texted based on these two cell lines (Appendix A). The best particle size for future applications is about 25.95 nm to maintain a greater hyperthermia effect, and a lower DOX molecule concentration could be achieved by using our designed carriers.

## 4. Conclusions

In summary, we developed a multifunctional Fe_3_O_4_@Au NP material as the molecule delivery system. Various size distribution of NPs conjugated with specific DNA sequence can load various amounts of DOX (as the anticancer) molecules. Due to the difference in the surface area, particle number, and steric hindrance effect, the number of DNA sequences bound varied with the core-shell nanoparticle size. DOX molecules can be controllably released from hyperthermia effect by means of HFMF triggering. The Au nano-shell caused a shielding effect on magnetic effect, which led to different SLP values and resulted in a 30−40% greater release than diffusion when applying HFMF for 20 min in a molecule-carrier system. The cytotoxicity and cancer treatment capability were examined using an in vitro cell viability test and showed lower cytotoxicity with higher DNA binding concentrations. However, the cell survival ratio decreased to 80% due to diffusion release from the molecule-loaded carriers. The application of HFMF resulted in the lowest dosage and could further decrease the side effects of both chemotherapy and hyperthermia. Fe_3_O_4_@Au core-shell NPs may be regarded as a multifunctional nanocarrier for the future theranostics of cancer.

## Data Availability

Data sharing not applicable.

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
