# Peer review of "The Multifunctionally Graded System for a Controlled Size Effect on Iron Oxide–Gold Based Core-Shell Nanoparticles"

_nanomaterials, 2021, doi:10.3390/nano11071695_

Round 1
Reviewer 1 Report
Dear Authors,
Your manuscript is interesting and topical. I propose to publish the manuscript.
One remark, I suggest to enlarge the scale's values and text on Figures.
Best wishes
Author Response
Thanks for your suggestion.
Reviewer 2 Report
The manuscript entitled “The multifunctionally graded system for a controlled size effect on iron oxide-gold based core-shell nanoparticles” by Bo-Wei Du et al describes the preparation, characterization of magnetic nanoparticles of different size, trying to correlate their sizes with their properties, namely binding and release of the drug DOxorubicin and its anti-tumor activity in vitro on two cell lines.
Some data are interesting, however, the authors use inappropriate terms, which makes difficult to fully and easily understand the concepts and the results described. Also, the authors should better explain the apparently discrepant results of Fig. 4 a-d, from which the reader does not see the expected dose-response relationship.
Another key point, which is not clearly explained is relative to the results on MCF-7 cells (Fig. 6) Why the difference between Hela cells and MCF-7 (Fig. 6 b and c)? Experiments reported in Fig 6 a and b do not show great differences when using nanoparticles of different sizes. Moreover, no statistical analysis with standard deviations are reported for the experiments reported in Fig 6d. How many times were the experiments with cells performed? What was the amount of nanoparticles used on cells? It is difficult to find the data, if they are present. The same for other quantitative data (see beyond). If the amounts of DOxorubicin were equalized for the different treatments in Fig. 6a (HeLa), the conclusions are that there is not a great difference between pure DOxo and nanoparticles loaded with DOxo + HFMF. In the case of MCF-7 (Fig. 6d) indeed it is clear the combined effect of chemotherapy and HFMF, if the same amount of pure DOxo and Doxo coupled on nanoparticles is used. The relationship between nanoparticle size and activity is difficult to be appreciated on the basis of the data.
For these reasons the manuscript cannot be accepted in the present form.
Details:
Page 2, line 72: modified is not an appropriate term, it should be substituted by coupled or something similar
Page 4, line 136: The acronym TMAOH should be reported in the MM
Page 4, lines 144-145. I am surprised that uncoated magnetic NPs can be removed by simple centrifugation. Do Au-coated MNPs display such a different weight?
Page 5, line 170 and then also line 208: HFMF: more details should be given: Hz used, …
Page 5, line 190. The term modified is not appropriate; maybe incubated/reacted
Page 5, line 190. Which concentration of Tween 20?
Page 5, line 205: subculture should be substituted with detached
Page 6, line 219: subcultured should be substituted with detached
Page 6, lines 223-228. This description is rather confused. It must be re-written completely to be understood. The basis for targeting is not clear at all.
The results in Fig 4 appear to be contradictory. They should be better explained.
At the end of the manuscript the authors report a single experiment on MCF-7 cells, which gave a significantly different result in respect to the ones observed on HeLa cells. It is not clear at all why this result was obtained. It is only written that the ds-DNA has specificity for MCF-7. However, more details about this point must be added and discussed.
Author Response
Thanks for your suggestion.
To the reviewer 2:
We acknowledge your positive comments and suggestions, which are valuable in improving the quality of our manuscript. We have changed all the unproper terms and added the details about HFMF to the Revised Manuscript. Furthermore, the more explanation has provided in the Revised Manuscript. We revised our manuscript in accordance with your instructive guidance. The Revised Manuscript has been a significant improvement than previous submission. As the editor asks us to reply reviewer comment before June 6, we try our best to meet the deadline. Thank you very much.
Your question I
Page 2, line 72: modified is not an appropriate term, it should be substituted by coupled or something similar.
Our response
√Based on reviewer’s comments, we have changed an appropriate term in the Revised Manuscript. Thanks for your helpful suggestion.
Your question II
Page 4, line 136: The acronym TMAOH should be reported in the MM.
Our response
√Based on the reviewer’s comments, the full name of TMAOH is shown in the Materials and Methods of the Revised Manuscript.
Your question III
Page 4, lines 144-145. I am surprised that uncoated magnetic NPs can be removed by simple centrifugation. Do Au-coated MNPs display such a different weight?
Our response
√The coating Fe3O4 nanoparticles have different weight with the uncoated one, and the final products can be isolated by centrifugation process. This phenomenon has been widely reported by many researchers (please refer to: (1) Synthesis of Fe oxide core/Au shell nanoparticles by iterative hydroxylamine seeding. Nano Letters 4.4 (2004): 719-723.; (2) Bio-modified Fe3O4 core/Au shell nanoparticles for targeting and multimodal imaging of cancer cells. Journal of materials chemistry 22.2 (2012): 470-477; (3) Continuous generation of hydroxyl radicals for highly efficient elimination of chlorophenols and phenols catalyzed by heterogeneous Fenton-like catalysts yolk/shell Pd@Fe3O4@metal organic frameworks. Journal of hazardous materials 346 (2018): 174-183; (4) Co-adsorption of gaseous benzene, toluene, ethylbenzene, m-xylene (BTEX) and SO2 on recyclable Fe3O4 nanoparticles at 0-101% relative humidities. Journal of Environmental Sciences 31 (2015): 164-174; (5) Amino acid coated superparamagnetic iron oxide nanoparticles for biomedical applications through a novel efficient preparation method. Journal of Cluster Science 28.3 (2017): 1259-1271 and (6) Catalano, Enrico. "In vitro biological validation and cytocompatibility evaluation of hydrogel iron-oxide nanoparticles." AIP Conference Proceedings. AIP Publishing LLC 1873 (2017): 020011). Hopefully we have addressed your concerns. Thank you very much.
Your question IV
Page 5, line 170 and then also line 208: HFMF: more details should be given: Hz used, …
Our response
√Based on reviewer’s concern, we have added the details about HFMF in the section 2.7 in the Materials and Methods of the Revised Manuscript. Thank you for your detailed advices.
Your question V
Page 5, line 190. The term modified is not appropriate; maybe incubated/reacted.
Our response
√Based on reviewer’s comments, we have changed an appropriate term in the Revised Manuscript. Thanks for your helpful suggestion.
Your question VI
Page 5, line 190. Which concentration of Tween 20?
Our response
√The 0.05% Tween 20 as the protection agent for Fe3O4@Au NPs was prepared in our experiment. As shown in Figure 6a, the HeLa cell viability after being treated with Fe3O4@Au NPs protected by Tween 20 was also tested in our manuscript. And we have provided the concentration value of Tween 20 in the Revised Manuscript. Thank you very much for your kindly suggestion.
Your question VII
Page 5, line 205: subculture should be substituted with detached.
Our response
√Based on reviewer’s comments, we have changed an appropriate term in the Revised Manuscript. Thanks for your helpful suggestion.
Your question VIII
Page 6, line 219: subcultured should be substituted with detached.
Our response
√Based on reviewer’s comments, we have changed an appropriate term in the Revised Manuscript. Thanks for your helpful suggestion.
Your question IX
Page 6, lines 223-228. This description is rather confused. It must be re-written completely to be understood. The basis for targeting is not clear at all.
Our response
√Based on reviewer’s comments, the Page6, line 223-228 in Section 2.10 have been clarified in Materials and Methods. Thanks for your helpful suggestion.
Your question X
The results in Fig 4 appear to be contradictory. They should be better explained.
Our response
√Based on reviewer’s concern, we have presented the explanation of Figure 4 properly in the Results and discussion of the Revised Manuscript. Thank you for your helpful advice.
Your question XI
At the end of the manuscript the authors report a single experiment on MCF-7 cells, which gave a significantly different result in respect to the ones observed on HeLa cells. It is not clear at all why this result was obtained. It is only written that the ds-DNA has specificity for MCF-7. However, more details about this point must be added and discussed.
Our response
√The MCF-7 breast cancer cell was chosen as a targeting cell due to its specific binding ability with the aptamer sequences (please refer to: Selective collection and detection of MCF-7 breast cancer cells using aptamer-functionalized magnetic beads and quantum dots based nano-bio-probes. Analytica chimica acta 788 (2013): 135-140.), which the HeLa cells do not have. Because the specific binding ability of MCF-7 with aptamer, and also as we explained in our article, on Page 2, line 87-90: “Because of the specific surface features of the Au nano-shell, the specially designed DNA sequence could be coupled with thiol bonding; subsequently, because CG-rich fragments carried DOX as the anticancer drug, both drug loading and specific targeting could be achieved ”. This indicated that the DOX which is binding on the surface of Fe3O4@Au NPs, as an antidrug, can also response to the target cells, which is the MCF-7 cell line. Furthermore, our target DNA is a CG-rich DNA which is used as a linker to carried the drug with core-shell nanoparticle.
In addition, on Page 5, line 242-243 and Page 8-9, line 419-421, as we mentioned: “Before washing with PBS buffer, all plates were placed in an incubator for 4 hr for targeting.” and “An aptamer sequence that had been demonstrated to have a specific binding ability with MCF-7 breast cancer cells was chosen, and HeLa cells were used as the control group”. The nanoparticles would stay on MCF-7 even after washed off by PBS, because the specific binding ability of the aptamer sequence with MCF-7, which the drug along with Fe3O4@Au NPs would be washed from HeLa cells. On the other hand, the cell viability of HeLa cells (Figure 6a) without washed by PBS demonstrated the lower cell viability as compared to the cell viability washed by PBS (Figure 6b). Hopefully we have addressed your concerns. Thank you very much.
Reviewer 3 Report
The article is of interest but it could be convenient modify part of the text:
- introduction. References 24 to 33 are not fully related with the topic, they could be removed. The last paragraph seems an abstract, please rewrite.
- Scheme 1 is not clear and the component are not found in the image
- Figura 3a the inset is not clear and is not explained in the figure caption.
Author Response
Thanks for your suggestion. Please refer to the file for our reply.

Reviewer 4 Report
In my opinion, the paper is intersting but lacks data to support multifunctionality.
To demonstrate multifunctionality Authors need to study the intracellular DOX fluorescence, T1 and T2 relaivity.
A control group of normal cells could be included.
Authors should disuss the benefits of using HAuCl4 solution rather than gold nanoseeds for functionalization.
The DOX concentration used should be indicated. DOX should be used at IC50 for each cell line. IC50 should be determined for all groups on both cell lines. Moreover, was the quantity of NP used for treatments normalized to DOX loading?
Gold thickness should be provided for all NPs.
Figure 6A should be improved as it is hard to distinguish columns.
Authors also need to demonstrate DOX release cycles.
Author Response

(The authors gave the same response as above.)

Round 2
Reviewer 2 Report
see attachment

Reviewer 4 Report
Authors adressed most comments so I do recommend acceptance of the revised paper.
Round 3
Reviewer 2 Report
I wait for the experiments I asked in the second revision and for that I wrote clearly to the Editor